# Deep imitation learning for molecular inverse problems

**Eric Jonas**
Department of Computer Science
University of Chicago
ericj@uchicago.edu

## Abstract

Many measurement modalities arise from well-understood physical processes and result in information-rich but difficult-to-interpret data. Much of this data still requires laborious human interpretation. This is the case in nuclear magnetic resonance (NMR) spectroscopy, where the observed spectrum of a molecule provides a distinguishing fingerprint of its bond structure. Here we solve the resulting inverse problem: given a molecular formula and a spectrum, can we infer the chemical structure? We show for a wide variety of molecules we can quickly compute the correct molecular structure, and can detect with reasonable certainty when our method fails. We treat this as a problem of graph-structured prediction where, armed with per-vertex information on a subset of the vertices, we infer the edges and edge types. We frame the problem as a Markov decision process (MDP) and incrementally construct molecules one bond at a time, training a deep neural network via imitation learning, where we learn to imitate a subisomorphic oracle which knows which remaining bonds are correct. Our method is fast, accurate, and is the first among recent chemical-graph generation approaches to exploit per-vertex information and generate graphs with vertex constraints. Our method points the way towards automation of molecular structure identification and active learning for spectroscopy.

## 1 Introduction

Understanding the molecular structure of unknown and novel substances is a long-standing problem in chemistry, and is frequently addressed via spectroscopic techniques, which measure how materials interact with electromagnetic radiation. One type of spectroscopy, nuclear magnetic resonance spectroscopy (NMR) measures properties about individual nuclei in the molecule. This information can be used to determine the molecular structure, and indeed this is a common, if laborious, exercise taught to organic chemistry students.

Although not usually conceived of as such, this is actually an inverse problem. Inverse problems are problems where we attempt to invert a known forward model $y = f(x)$ to make inferences about the unobserved $x$ from measurements $y$. Inverse problems are at the heart of many important measurement modalities, including computational photography [31], medical imaging [5], and microscopy [22]. In these cases, the measurement model is often linear, that is $f(x) = Ax$, and the recovered $x$ is a vector in a high-dimensional vector space (see [24] for a review ). Crucially, $A$ is both linear and *well-known*, as a result of extensive hardware engineering and calibration.

Nonlinear inverse problems have a *non-linear* forward model $f$, and *structured inverse problems* impose additional structural constraints on the recovered $x$, like being a graph or a permutation matrix. Spectroscopy is an example of this sort of problem, where we wish to recover the graph structure of a given molecule from its spectroscopic measurements. Yet we still know $f$ very well, in the case

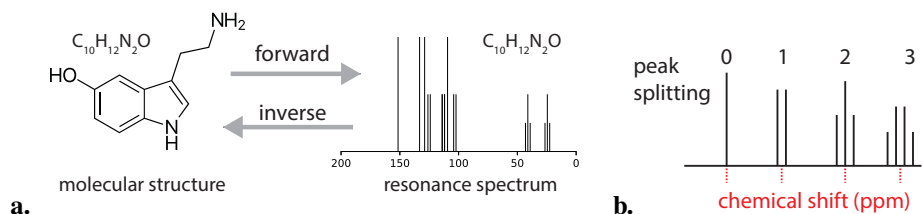

Figure 1: **a.** The **forward problem** is how to compute the spectrum of a molecule (right) given its structure (left). Spectroscopists seek to solve the corresponding **inverse problem**, working backward from spectra towards the generating structure. **b.** Various properties measured in a spectrum, including the chemical shift value and the degree to which each peak is split into multiplets.

of NMR due to decades of expertise in quantum physics and computational chemistry, especially techniques like Density Functional Theory (DFT) [23, 10].

Here the inverse problem we tackle is structured as follows: Given a collection of per-atom (vertex) measurements, can we recover the molecular structure (labeled edges) ? Note we seek a single, correct graph – unlike recent work in using ML techniques for molecules, here there is one true, correct edge set (up to isomorphism) that we are attempting to find. We formulate the problem as a Markov decision process, and use imitation learning to train a deep neural network to sequentially place new bonds until the molecule is complete. The availability of a fast approximate forward model[1] $f$ enables us to rapidly check if our recovered structure matches experimental data, allowing us more confidence in the veracity of our recovered structure.

In this paper, we briefly review the relevant graph-theoretic connections to chemistry and NMR spectroscopy, explain the sequential problem setup, our dataset, and how the resulting Markov decision process (MDP) can be solved via imitation learning. We evaluate on both synthetic and real data, showing that the resulting problem is invertible, is reasonably well-posed in the face of input data perturbations, and crucially performs well on experimentally-observed data for many molecules. We assess how improvements to both the approximate forward model $f$ and the inverse that we propose here can be made, and suggest avenues for future research.

## 1.1   Basic chemistry

We can recall from early education that a molecule consists of a collection of atoms of various elements, with certain types of "bonds" between those atoms. This corresponds to a following.

A molecule is

1. A collection of vertices (atoms) $V = \{v_i\}$, each vertex having a known color (that is, corresponding to a particular element) and maximum vertex degree, corresponding to the valence of that element.

2. A set of edges $E = \{e_{ij}\}$ between the vertices $(v_i, v_j) \in V$ each with an associated edge label $c$ corresponding to their *bond order*. The bond orders we are interested in are labeled $\{1, 1.5, 2, 3\}$ corresponding to single, aromatic, double, and triple bonds.

3. A graph $G = (V, E)$ which is connected, simple, and obeys a color-degree relationship by which the weights of the edges connected to a vertex $v_i$ almost always sum to a known value dependent on that vertices' color.

Our measurement technique, NMR spectroscopy, lets us observe various per-vertex properties, $P(v_i)$, which we can then featurize. Note we only observe $P(v_i)$ for a subset of nuclei, and will refer to the set of all of these observations as $P$.

We begin with a known molecular formula, such as $C_8H_{10}N_4O_2$, obtained via stoichiometric calculations or an alternative techniques such as mass spectroscopy. In our application of NMR spectroscopy,

there are two sources of per-vertex information measured (fig 1b) in an experiment: a unique per-nucleus resonance frequency, the *chemical shift*, that arises as a function of the local electronic environment, and *peak splitting*, which results from spin-spin coupling between a nucleus and its neighbors. In this work we focus on $^{13}$C spectroscopy and thus only observe these properties for the carbon atoms; all other nuclei yield no spectroscopic information. In this case, the splitting then reflects the number of adjacent hydrogens bonded to the carbon. Thus, our problem ultimately is as follows: **based upon a collection of per-vertex measurements we wish to know the label (if any) of the edge $e_{ij}$ between all vertex pairs** $(v_i, v_j)$.

## 2    Sequential molecule assembly via imitation learning

We formulate our problem as a Markov decision process (MDP), where we start with a known set of vertices $V$ and their observed properties $P$, and sequentially add bonds until the molecule is connected and all per-vertex constraints are met. Let $G_k = (V, P, E_k)$ represent the $k$th state of our model, with $k$ existing edges $E_k$, $k \in [0, \dots, K]$. We seek to learn the function $p(e_{k+1} = e_{i,j,c} | E_k, V, P)$ assigning a probability to each possible new edge between $v_i$ and $v_j$ (with label $c$) and in $G_{k+1}$.

To generate a single molecule (given spectral observations $P$ and per-vertex information $V$) we begin with an empty edge set $E_0$ and sequentially sample from $E_k \sim p(e_{i,j,c} | E_k, V, P)$ until we have placed the correct number of edges necessary to satisfy all the valence constraints. We call this resulting edge set (molecule) a *candidate structure*. For a given spectrum $P$, we can repeat this process N times, generating N (potentially different) candidate structures, $\{E_K^{(i)}\}_{i=1}^N$. We can then evaluate the quality of these candidate structures by measuring how close their predicted spectra $P_i = f(E_K^{(i)}$ are to the true observations $P$, that is,

$$E_{predicted} = \text{argmin}_i ||f(E_K^{(i)}) - P||_2$$

This is crucial – our sequential molecule generation process can produce a large number of candidates but the presence of the fast forward model can rapidly filter out incorrect answers.

### 2.1    Imitation learning

For any partially completed molecule $G_k = (V, P, E_k)$ we seek out the single correct edge set $E_K$. We can use our ability to efficiently compute graph subisomorphism as an oracle – at training time we compute which individual edges $e_{i,j,c}$ we could add to $E_k \cup \{e_{ijc}\}$ such that $G_{k+1}$ would still be subisomorphic to $G_K$ ( details of this subisomorphism calculation are provided in the appendix 6.2)

We generate a large number of candidate $(G_k, E_{k+1})$ training pairs, and fit a function approximator to $p(e_{i,j,c} | E_k, V_p)$. To generate these training pairs, we can take a known molecule $G_K$ with observed properties $P$ and delete $d$ edges randomly yielding state $G_{K-d}$ (see figure 3). Note this is a crucial difference between our problem and previous molecule generation work: at each point when assembling a molecule, there is a finite, known set of valid next edges, as we are seeking a single correct structure.

There are many ways to sample the number of deleted edges $d$ and a pathological choice may yield distributional shift issues, where sequentially-assembled molecules during the recovery phase might look significantly different from our training molecules. We did not observe this in practice, although adoption of distributional-shift techniques such as SEARN [6] or DAGGER [25] are worth exploration in the future. Here we sample $d$ such that the fraction of remaining edges is distributed as a mixture of uniform and beta distributions, $d \sim 0.2 \cdot \text{Unif}(0, 1) + 0.7 \cdot \text{Beta}(3, 3)$.

### 2.2    Dataset

Computing the spectrum from a given complete molecule $G$ requires computationally-expensive quantum-mechanical (DFT) calculations, and there is a paucity of high-quality NMR experimental data available. We leverage the authors' previous work on learning fast neural-network-based approximations for spectra to address this problem[**?** ]. Our original fast forward model was trained on a small number of experimentally-available spectra, and then that model was used to synthesize the spectra of a very large number of candidate molecular structures.

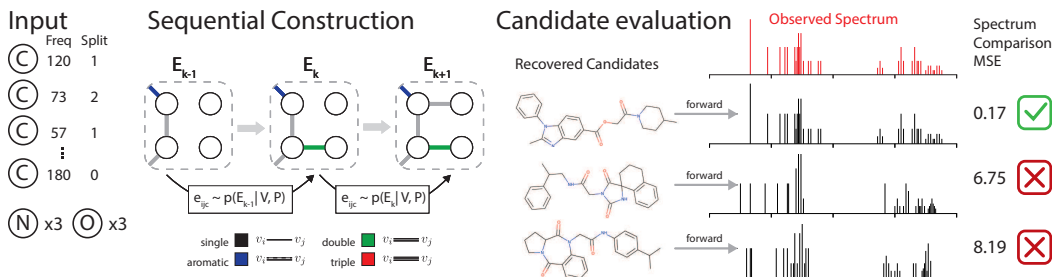

Figure 2: a.) Our input is a molecular formula indicating the number of atoms (vertices) of each element (color) along with per-vertex properties $P$ measured for a subset of those vertices (in this case, carbon nuclei). b.) We sequentially construct a molecule by sampling the next edge $(i, j)$ and edge label $c$ conditioned on the existing edge set as well as vertices $V$ and vertex properties $P$. c.) We end up with a sampled collection of candidate molecules, which we can then pass back through our forward model to compute their spectra, and validate against the true (observed) spectrum.

We obtain molecular structures consisting of H, C, O, and N atoms by downloading a large fraction of the public database PubChem []. We select molecules with no more than 64 atoms total and at most 32 heavy atoms (C,O,and N). We preprocess this dataset to eliminate excessively small (<5 atoms) rings and excessively large rings (> 8 atoms), as well as radicals, as they were absent in the initial dataset that trained our approximate forward model. We take care to ensure that no molecule is duplicated in our database, contaminating our train/test split. This leaves us with 1,268,461 molecules, of which we train on 819,200 and draw the test set from the remainder.

## 2.3 Network

Our network assigns a probability to each possible edge and edge-label given the current partially-competed graph, $p(e_{i,j,c}|E_k, V, P)$. The network itself is a cascaded series of layers which combine per-edge and per-vertex information, sometimes called a "relational network" and nicely summarized in [2]. Each layer $k$ of the network transforms per-vertex $v^{(k)}$ and per-edge features $e^{(k)}$ into per-vertex $v^{(k+1)}$ and per-edge $e^{(k+1)}$ output features, see figure 3. Crucially the network combines pairwise vector features with the associated edge features $e'_{ij} = \phi_e(e_{ij} + v'_i + v'_j)$ when computing the next edge feature, and computes an aggregate of all edges associated with a vertex $v_i^e = \max_i e'_{ij}$. We adopt a residual structure as we find it easier to train, and our $\phi_r$ perform batch normalization before ReLu operations, We found that a recurrent GRU cell was the easiest to train efficiently when combining input per-vertex features with the aggregated edge features $G(v_i^e, v_i^{(k)})$. Note our network at no point depends on the ordering of the input per-vertex or per-edge features, thus maintaining permutation-invariance.

The first layer takes in per-vertex properties including element type, typical valence, and others (see appendix 6.1) as well as the experimentally-observed per-vertex measurements for the relevant nuclei, including one-hot-encoded (binned) chemical shift value and peak splitting. We also featurize the current state of the partially-assembled molecule as a one-hot-encoded adjacency matrix, where $e_{i,j,c}^{\text{in}} = 1$ if there is an edge labeled $c$ between $v_i$ and $v_j$ and 0 otherwise.

We cascade 32 relational layers and an internal feature dimensionality of 256, discarding the final layer's per-vertex information. Each output edge is passed through a final 1024-dimensional fully-connected layer to reduce its dimensionality down to 4 for output outputting a one-hot-encoded edge weight for each of the four possible edge labels. We normalize the resulting outputs to give us the final $p(e_{i,j,c}|E_k, V, P)$.

We train using binary cross-entropy loss between the allowed possible next edges and the predicted edges of the network. We use Adam with a learning rate of $10^{-4}$ and train the network for 100 hours on an nVidia v100 GPU.

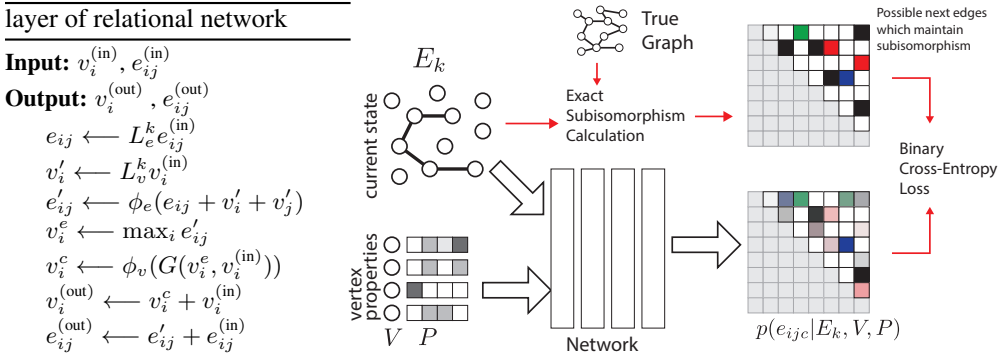

Figure 3: Each layer $k$ of the network transforms per-vertex $v^{(\text{in})}$ and per-edge features $e^{(\text{in})}$ into per-vertex $v^{(\text{out})}$ and per-edge $e^{(\text{out})}$ output features. At train time we take true graphs, randomly delete a subset of edges, and exactly compute which single edges could be added back into the graph and maintain subisomorphism. We minimize the binary cross-entropy loss between the output of our network and this matrix of possible next edges.

## 2.4 Related work

Early work on function approximation and machine learning techniques for inverse problems, including neural networks [16], focused primarily on one-dimensional problems. More recent work has focused on enhancing and accelerating various sparse linear recovery problems, starting with [9]. These approaches can show superior performance to traditional reconstruction and inversion methods [29], but a large fraction of this may be due to learning better data-driven regularization. [24].

Deep learning methods for graphs have attracted considerable interest over the past several years, beginning with graph convolutional kernels [15] and continuing to this day (see [2] for a good unifying view). Many of these advances have been applied to chemical problems, including trying to accurately estimate whole molecule properties [28], find latent embeddings of molecules in continuous spaces [8], and optimize molecules to have particular scalar properties [11]. These approaches have focused on exploiting the latent space recovered by GANs [7] and variational autoencoders [26]. Crucially these molecule generation approaches have largely focused on optimizing a single scalar property by optimizing in a learned latent space, which had made constraints like the precise number and elements of atoms difficult.

Other approaches have attempted to sequentially construct graphs piecemeal [21, 30] using RNNs and deep Q-learning, but here we have an oracle policy that is *known to be correct*: our ability to explicitly calculate subisomorphism between $E_{k+1} = e_{ijc} \cup E_k$ and the final $G_K$. This lets us reduce this to learning to imitate this oracle, a substantially easier problem.

Recent work on structured prediction via imitation learning, sometimes termed *learning to search*[6], is a strong inspiration for this work, (see [3, 4] for recent results with a good review). Our work extends these approaches to dense connected graphs via an explicit subisomorphism oracle.

Finally, the use of machine learning methods for chemical inverse problems is historic, especially with regards to other types of spectroscopy, such as mass spec. One of the first expert systems, DENDRAL, [20] was conceived in the early 1960s to elucidate chemical structures from mass spectroscopy data, which fractures elements and measures the charge/mass ratio of the subsequent fragments. Today, commercial software exists [1] to perform chemical structure determination from two-dimensional NMR data, a richer but substantially slower form of NMR data acquisition, which measures **both** chemical shift values and scalar coupling constants between nuclei, which can directly inform bond structure. Our focus on 1-D NMR has the potential to dramatically accelerate this structure discovery process.

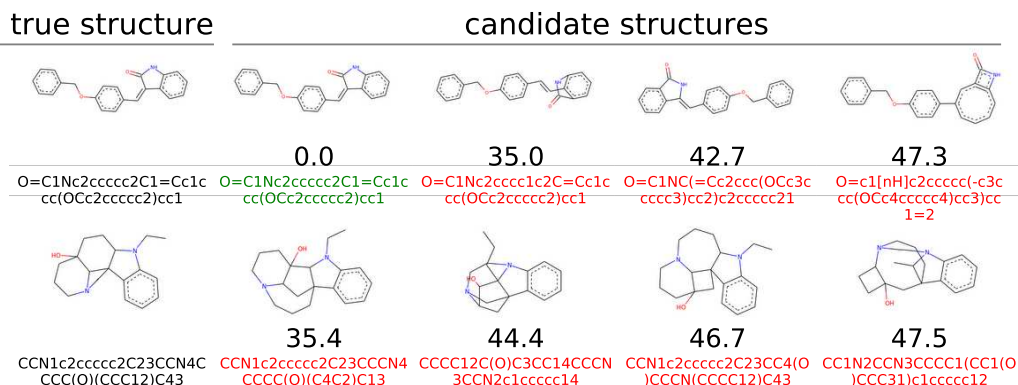

| true structure | candidate structures | | | |
|---|---|---|---|---|
| O=C1Nc2ccccc2C1=Cc1c cc(OCc2ccccc2)cc1 | 0.0<br>O=C1Nc2ccccc2C1=Cc1c cc(OCc2ccccc2)cc1 | 35.0<br>O=C1Nc2cccc1c2C=Cc1c cc(OCc2ccccc2)cc1 | 42.7<br>O=C1NC(=Cc2ccc(OCc3c cccc3)cc2)c2ccccc21 | 47.3<br>O=c1[nH]c2cccc(-c3c cc(OCc4ccccc4)cc3)cc 1=2 |
| CCN1c2ccccc2C23CCN4C CCC(O)(CCC12)C43 | 35.4<br>CCN1c2ccccc2C23CCCN4 CCCC(O)(C4C2)C13 | 44.4<br>CCCC12C(O)C3CC14CCCN 3CCN2c1ccccc14 | 46.7<br>CCN1c2ccccc2C23CC4(O )CCCN(CCCC12)C43 | 47.5<br>CC1N2CCN3CCCC1(CC1(O )CCC31)c1ccccc12 |

Figure 4: Example recovered structures. The left-most column is the true structure and associated SMILEs string, and the right are the candidate structures produced by our method, ranked in order of spectral reconstruction error (number below). Color indicates correct SMILEs string. The top row shows a molecule for which the correct structure was recovered, and the bottom row is an example of a failure to identify a structure.

## 3  Evaluation

Our molecule samples candidate structures for a given set of observations using our trained policy, and then we check each of those structures' agreement with the observed data using our fast forward model. We call this error in reconstruction the *spectral reconstruction error* for candidate edge set $E_K^{(i)}$

$$S_e = ||f(E_K^{(i)}) - P||_2$$

While training we attempt to generate graphs with exact isomorphism – that is, the recovered graph is isomorphic to the observed graph, including all per-vertex properties. This is excessively stringent for evaluation, as the experimental error for chemical shift values can be as high as 1 ppm, thus experimentally a carbon with a shift of $115.2$ and a carbon with a shift of $115.9$ are are indistinguishable. Thus, we relax our definition and conclude two molecules are correct if they agree, irrespective of exact shift agreement, according to a *canonicalized string representation*. Here we use SMILES strings [27], which are canonical string representations of molecular structure with extensive use in the literature. SMILES strings are a more realistic definition of molecular equivalence. We use the SMILES generation code in RDKit [19], a popular Python-based chemistry package.

### 3.1  Inversion, identifiability, noise sensitivity

First we assess our ability to simply recover the correct molecular structure given observed data, operating entirely on simulated data, using the exact spectra produced by our forward model $f$. This noise-free case represents the best case for our model, and lets us assess the degree to which our system works at all – that is, is the combination of chemical shift data and peak splitting data sufficient to uniquely recover molecular structure?

For each spectrum in the test set our model generates 128 candidate structures and measure the fraction which are SMILES-correct. In figure 4 we show example spectra and recovered molecules, both correct and incorrect (see appendix 6.3 for more). Since in this case we are using spectral observations computed by our forward model, the spectral reconstruction error should be exactly 0.0 for the correct structure, and all incorrect structures have substantially larger spectral errors when passed through the forward model.

For each test molecule, which fraction of the 128 generated candidate models are correct, irrespective of how well the forward model matches? This is a measure of how often we produce the correct structure, and in the noiseless case is the best case. Figure 5a shows the fraction of correct candidate structures sorted in decreasing order; we can see that 90.6% molecules in the test set have at least

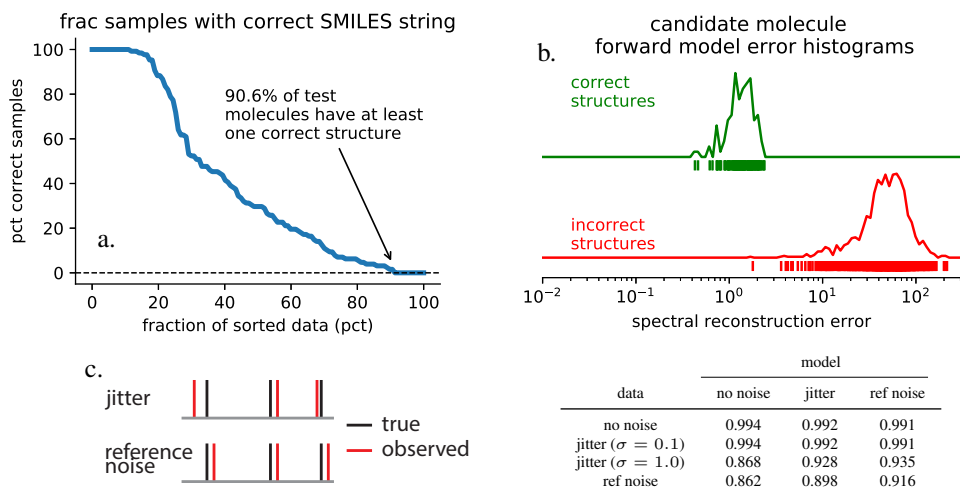

Figure 5: **Synthetic data: a.** Distribution of candidate molecules that are correct, before filtering, for the noiseless case. For 90.6% of the test molecules we generate at least one correct structure. **b.** Distribution of reconstruction errors for correct and incorrect candidate structures, showing a clear separation. **c.** Jitter adds random noise to each shift value independently, whereas reference noise shifts the entire spectrum by a random amount. We train models on each class of noise and evaluate on each class of model. The performance as measured by AUC minimally degrades when the model is trained with the equivalent noisy (augmented) data.

one generated structure that is correct. We then pass each of these candidate structures back through the forward model and evaluate the MSE between the observed spectrum for the true molecule and the spectra of the candidates. Figure 5b shows the distribution of reconstruction error for the correct structures and for the incorrect structures; a clear separation is visible, suggesting the existence of a threshold that can be used to reliably distinguish correct from incorrect structures.

## 3.2   Well-posedness

Inverse problems are often ill-posed in the sense of Hadamard [14], where the solution can vary wildly with slight perturbations in the input. To evaluate the ill-posedness of our model, again generate data with our forward model, but we perturb these spectra at evaluation time via with per-chemical-shift jitter and reference noise.

Per-shift jitter can arise in SNR-limited applications when precise measurement of the chemical shift peak is difficult, and we model here as the addition of Gaussian noise, training with $\sigma = 0.5$ and evaluating with $\sigma = \{0.1, 1.0\}$. Reference noise can arise when the reference frequency is mis-calibrated or mis-measured and manifests as an offset applied to all observed chemical shift values. Here we model this as a uniform shift on $[-2, 2]$ ppm.

As we have added noise to the observed data at evaluation time, in this case our spectral reconstruction error $S_e$ will never be zero – even the candidate structure with the lowest $S_e$ will not match the forward model exactly. We can set a threshold for this reconstruction error, below which we will simply choose not to suggest a solution – in this case our model is effectively saying "I don't know, none of these candidate structures fit well enough". For a given evaluation set of molecules, we can vary this threshold – when it is very low, we may only suggest solutions for a small number of molecules (but they are more likely to be correct). We can then vary this threshold over the range of values, and compute the resulting area under the curve (AUC). We use this threshold-based confidence approach throughout the remainder of the paper.

We train our model with no noise in the training data, and with training data augmented with just per-shift jitter, and with reference noise *and* per-shift jitter. We evaluate the resulting models on evaluation datasets with no noise, jitter at two values, and reference noise (figure 5c).

| evaluation data from | | | threshold accuracy | | | | |
|---|---|---|---|---|---|---|---|
| forward | inverse | AUC | 10% | 20% | 50% | top-1 | top-5 |
| train | train | 0.95 | 100.0% | 100.0% | 99.2% | 79.5% | 79.9% |
| | test | 0.88 | 100.0% | 99.0% | 96.9% | 71.3% | 74.6% |
| test | train | 0.91 | 100.0% | 100.0% | 97.6% | 60.3% | 63.3% |
| | test | 0.83 | 100.0% | 97.1% | 90.2% | 55.9% | 58.6% |

Table 1: **Experimental data:** Evaluation on varying degrees of unseen molecules. "Train" and "test" indicate whether the forward or inverse model was trained on the evaluation set or not. ("test", "test") represent molecules completely unseen at both training and test time, and reflect the most realistic scenario. "theshold accuracy" reflects the SMILES accuracy of the top n% of molecule-prediction matches, by spectra MSE. Top-1 and top-5 are average correctness across the entire dataset of the top-1 and top-5 candidate structures.

We can see from the table that training the presence of noise can substantially improve noise robustness, and reference artifacts are more damaging to our recovery AUC, even when we train the model with them. Thus while our model is robust to some classes of perturbations, the precise chemical shift values do matter. Systematic exploration of this sensitivity is an obvious next step for future work.

## 4   Experimental data

All previous results have been using spectra generated from our fast forward model. Now we compare with actual experimentally-measured spectra. We use NMRShiftDB [18] for all data. The original forward model was trained on a small number of experimentally-available spectra, and then that model was used to synthesize the spectra of a very large number of molecular structures from PubMed. This gives us four different classes of experimental molecules to evaluate: molecules from the train and test sets of the forward model, and molecules whose structures were used in training the inverse network or not.

We evaluate this performance in table 1. We measure the AUC as we vary the "reconstruction error" threshold, and look at the accuracy of the top 10%, 20%, and 50% molecules with the lowest reconstruction error.

For molecules whose structure was never seen by either the approximate forward model or the inverse model, we achieve an AUC of $0.83$ and a top-1 accuracy of $55.9\%$. We can select a $S_e$ threshold which gives a prediction for $50\%$ of the molecules, and in this case we recover the correct structure $90.2\%$ of the time.

We can use the other forward/inverse pairings to sanity check these results and understand the relative importance of molecular structure and chemical shift prediction between the forward and the inverse model. Unsurprisingly when the structure was known to both the forward and the inverse model, we achieve the best performance, even though the inverse model *never observed the empirical shifts*, only the simulated shifts from the forward model. AUC drops by 0.07 - 0.08 between train and test sets for the inverse model, regardless of whether or not the molecule was in forward model training or test. Similarly, AUC drops by 0.04-0.05 between forward model train and test, regardless of whether the inverse model had seen the molecule or not. This is suggestive of a consistent independent role for both forward model accuracy and inverse model performance, and suggests that improving either will ultimately yield improved reconstruction performance.

## 5   Discussion

Here we have demonstrated the capability to reconstruct entire molecular graphs from per-vertex spectroscopic information via deep imitation learning. This inverse problem is made viable by the on-line generation of an enormous amount of training data, enabled by the construction of a fast forward model to quickly compute these per-vertex properties, and efficient computation of exact graph subisomorphism.

Our approach is very general for molecular spectroscopy problems, but many steps and extensions remain before we can completely automate the process of chemical structure identification. While our approach gives us the ability to evaluate the quality of our candidate structures (by passing them back through the forward model and computing MSE), this approach is not perfect, and there are numerous avenues for improvement, both to expand the fraction of molecules we confidently predict and the resulting accuracy.

Fist, we began by focusing on molecules with the four most common organic elements (H, C, O, and N) but plan to extend our method to other elements. One challenge is that the accuracy of chemical shift prediction for NMR spectra is less accurate for other important organic elements like S,P,F, and Cl. Even best-in-class density functional theory calculations struggle to accurately compute *ab initio* chemical shift values for heavier atoms such as metals, and molecules with unusual charge distributions such as radicals [17].

Here we focus entirely on $^{13}$C NMR, but $^1$H NMR has a substantially higher signal-to-noise ratio (SNR) due to the near-100% natural abundance of the spin-active hydrogen isotope. However, the chemical shift range for hydrogen is substantially smaller ($\sim$ 9ppm) and much more variable. Nonetheless, a high-quality forward model approximation for hydrogen spectra would enable us to easily incorporate this information into our model, and would potentially improve performance.

Like the majority of machine learning approaches to chemistry, by focusing entirely on topology we are ignoring substantial geometric effects, both stereochemical and conformational. As different stereoisomers can have radically different biological properties, correctly identifying steroisomerism is an important challenge, especially in natural products chemistry. Incorporating this explicit geometry into both our forward model approximation and our inverse recovery is an important avenue for future work.

More complex forms of NMR are capable of elucidating more elements of the coupled spin system created by a molecule, giving additional information about atomic bonding. As noted above, this is the basis for the (substantially slower) technique present in 2D NMR. Incorporating this style of data into our model is relatively straight-forward assuming we can develop a forward model that accurately predicts the parameters for the full spin Hamiltonian, and has the potential to extend our performance to substantially-larger molecules.

Finally, we see tremendous potential in using techniques like the one outlined here to make the spectroscopic process *active*. As the NMR spectrometer is fully programmable, it may be possible to use the list of uncertain candidate structures to compute successive follow-on experiments to more confidently identify the true structure. This style of active instrumentation will require the existence of very rapid forward and inverse models, of which we believe models like those presented in this work are an important first step.

## Footnotes

[1]We use this model throughout the rest of the paper and will treat it as "correct", thus will frequently drop the "approximate" designation.

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
