[Supplementary Material]

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

| layer of relational network |
| --- |

**Input:** $v_i^{(\mathrm{in})}, e_{ij}^{(\mathrm{in})}$

**Output:** $v_i^{(\mathrm{out})}, e_{ij}^{(\mathrm{out})}$

$$e_{ij} \longleftarrow L_e^k e_{ij}^{(\mathrm{in})}$$
$$v_i' \longleftarrow L_v^k v_i^{(\mathrm{in})}$$
$$e_{ij}' \longleftarrow \phi_e(e_{ij} + v_i' + v_j')$$
$$v_i^e \longleftarrow \max_i e_{ij}'$$
$$v_i^c \longleftarrow \phi_v(G(v_i^e, v_i^{(\mathrm{in})}))$$
$$v_i^{(\mathrm{out})} \longleftarrow v_i^c + v_i^{(\mathrm{in})}$$
$$e_{ij}^{(\mathrm{out})} \longleftarrow e_{ij}' + e_{ij}^{(\mathrm{in})}$$

Figure 3: Each layer $k$ of the network transforms per-vertex $v^{(\mathrm{in})}$ and per-edge features $e^{(\mathrm{in})}$ into per-vertex $v^{(\mathrm{out})}$ and per-edge $e^{(\mathrm{out})}$ output features. At train time we take true graphs, randomly delete a subset of edges, and exactly compute which single edges could be added back into the graph and maintain subisomorphism. We minimize the binary cross-entropy loss between the output of our network and this matrix of possible next edges.

## 2.4  Related work

Early work on function approximation and machine learning techniques for inverse problems, including neural networks [16], focused primarily on one-dimensional problems. More recent work has focused on enhancing and accelerating various sparse linear recovery problems, starting with [9]. These approaches can show superior performance to traditional reconstruction and inversion methods [29], but a large fraction of this may be due to learning better data-driven regularization. [24].

Deep learning methods for graphs have attracted considerable interest over the past several years, beginning with graph convolutional kernels [15] and continuing to this day (see [2] for a good unifying view). Many of these advances have been applied to chemical problems, including trying to accurately estimate whole molecule properties [28], find latent embeddings of molecules in continuous spaces [8], and optimize molecules to have particular scalar properties [11]. These approaches have focused on exploiting the latent space recovered by GANs [7] and variational autoencoders [26]. Crucially these molecule generation approaches have largely focused on optimizing a single scalar property by optimizing in a learned latent space, which had made constraints like the precise number and elements of atoms difficult.

Other approaches have attempted to sequentially construct graphs piecemeal [21, 30] using RNNs and deep Q-learning, but here we have an oracle policy that is *known to be correct*: our ability to explicitly calculate subisomorphism between $E_{k+1} = e_{ijc} \cup E_k$ and the final $G_K$. This lets us reduce this to learning to imitate this oracle, a substantially easier problem.

Recent work on structured prediction via imitation learning, sometimes termed *learning to search*[6], is a strong inspiration for this work, (see [3, 4] for recent results with a good review). Our work extends these approaches to dense connected graphs via an explicit subisomorphism oracle.

Finally, the use of machine learning methods for chemical inverse problems is historic, especially with regards to other types of spectroscopy, such as mass spec. One of the first expert systems, DENDRAL, [20] was conceived in the early 1960s to elucidate chemical structures from mass spectroscopy data, which fractures elements and measures the charge/mass ratio of the subsequent fragments. Today, commercial software exists [1] to perform chemical structure determination from two-dimensional NMR data, a richer but substantially slower form of NMR data acquisition, which measures **both** chemical shift values and scalar coupling constants between nuclei, which can directly inform bond structure. Our focus on 1-D NMR has the potential to dramatically accelerate this structure discovery process.

| true structure | candidate structures |
|---|---|

| | 0.0 | 35.0 | 42.7 | 47.3 |
|---|---|---|---|---|
| O=C1Nc2ccccc2C1=Cc1c cc(OCc2ccccc2)cc1 | O=C1Nc2ccccc2C1=Cc1c cc(OCc2ccccc2)cc1 | O=C1Nc2cccc1c2C=Cc1c cc(OCc2ccccc2)cc1 | O=C1NC(=Cc2ccc(OCc3c cccc3)cc2)c2ccccc21 |

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

# 6  Appendix

## 6.1  Network details

For each input molecule in the training set we compute features that depend on the vertices and their assigned properties (table 2), per-vertex features which are informative about the current edge state $E_k$ of the partially-assembled molecule $G_k$ (table 3), and per-edge current edge state features $E_k$ (table 4)

Table 2: per-vertex features state-independent features

| attribute | description | dimensionality |
|---|---|---|
| atomic number | one-hot encoded (C, N, O) | 3 |
| default valence | one-hot-encoded (1-3) | 3 |
| spectrum | binarized, 64-bins, 0-220 (see below) | 64 |
| observed peak splitting | one-hot encoded (0-3) | 4 |

We one-hot encode the observed chemical shift value for the nuclei where it is measured into a series of discrete bins (64) over a range of 0-220 ppm. Rather than merely discretizing the shift value, we discretize a Gaussian with $\sigma = 2$ allowing for information to be shared between bins. Early experiments revealed this to improve performance.

Table 3: per-vertex state-dependent features

| attribute | description | dimensionality |
|---|---|---|
| current degree | summed effective edge weight | 1 |
| remaining degree | remaining effective edge weight | 1 |
| remaining degree | remaining degree one-hot encoded [-1.0, 0.0, 1.0, 1.5, 2.0, 2.5, 3.0, 4.0] | 8 |

Table 4: per-edge state-dependent features

| attribute | description | dimensionality |
|---|---|---|
| edge label | edge label as weight | 1 |
| edge presence | is there an edge here | 1 |
| edge label | one-hot-encoded edge label [0, 1, 1.5, 2, 3] | 5 |

## 6.2 Calculation of graph subisomorphism

Efficient calculation of graph subisomorphism is a computationally challenging problem, with no known solution for all cases. Fortunately several existing high-quality libraries exist which take advantage of reasonable heuristics and work well in practice. We use the recently-developed VF2++ algorithm and implementation [13] which we found to be nearly twice as fast as existing libraries with substantially better performance on outlier cases. VF2++ only supports labeled vertices, not edges, so we transform each input graph with labeled edges into a bipartite graph with labeled edge nodes, and subsequently compute subisomorphism. Note that we perform a subisomorphism calculation for every candidate possible next edge in every molecule in a minibatch. At our size and problem complexity, on average computing the full set of next edges for a molecule takes roughly 10 ms, and with appropriate threading can be done online and does not rate-limit the training process.

## 6.3 Example results

Figure 6: Left: correctly recovered structures. The left-most column is the true structure and associated SMILEs string, and the right are the candidate structures produced by our method, ranked in order of spectral reconstruction error.

Figure 7: Left: incorrectly recovered structures. The left-most column is the true structure and associated SMILEs string, and the right are the candidate structures produced by our method, ranked in order of spectral reconstruction error. Note that none of the indicated structures has a low spectral agreement.

## 6.4 Brief overview of the forward model

We make extensive use of a fast forward model described in [12], where we use a small collection of empirically-measured spectra to train a model. Here we briefly summarize those results.

We start user-contributed measured experimental spectra for 32538 molecules from NMRShiftDB2 and train a edge-sensitive (that is, bond-order-aware) graph neural network to encode per-vertex properties, which we then use several feedforward (linear) layers to predict both a mean and a variance for the requested chemical shift. We train with a variational loss which approximately fits a small Gaussian to the observed data, parameterized by mean and variance. We found this had the added benefit of making it easier to fit/ignore the outliers in our source-database, of which there were many due to its user-contributed nature.

Ultimately, we use this model to generate the estimated $^{13}$C chemical shifts for our input data. We only make use of the predicted shift mean – future work could take the confidence interval into account for more accurate data augmentation (beyond our simple Gaussian noise model).