[Reviews · NeurIPS 2019]

Reviewer 1



The model doesn't seem to generalize very well, especially the AUC for unseen structures is low.

Reviewer 2



The paper is clearly written and motivates the interesting application of finding molecular structures given a spectrum well. The structure of the ms could be improved, since there are some distracting jumps between method, experiments and related work. In particular, the evaluation (Sec 4) could be described in more detail and can be confusing at the first reading. For example, the threshold was only mentioned once before and it could be stated again, that it applies to the spectrum, not the geometry. Here, the paper could also benefit from giving an overview of the training and evaluation procedure, e.g. in a flow chart. The components of the proposed approach - an autoregressive graph model and a predictor of molecular properties (which is unfortunately not described in detail to keep double-blindness) - are known from the literature, but are combined in a new way to solve the structured inverse problem. The presented approach could serve as a blueprint for other molecular inverse design problems, in particular concerning the efficient use of simulated and experimental data.

Reviewer 3



The fast forward model to generate the simulated spectra is not clear, which makes hard to understand the general picture and the theoretical results. This should be described in a clear and extended manner in the supplementary material. Ideally the code should be provided. AUC not defined. In Related work, a summary of references is given but neither substantial analysis nor hard numbers to compare are provided. Line 108-109: Elaborate in the meaning of this sentence to make it clearer, e.g. what is Beta? Missing labels in Fig. 5. Line 203: Forgot to remove note? Did the authors verify that 90.6% was the correct value? Line 222-223: Inverse problems are, in general, a difficult topic and therefore in any application systematic analysis are mandatory. It is not clear how the stability analysis performed really guaranties the robustness of the method. I would suggest a more extended and clear study should be performed.

[Author Response · NeurIPS 2019]

We wish to thank the reviewers for their insightful and constructive reviews. We will attempt to address some of the raised questions and points below, and indicate several places where we will update both the camera-ready manuscript and the supplemental material.

First, reviewer three is absolutely correct that inverse problems are hard and require rigorous and careful experimentation to validate new approaches. The literature often refers to "inverse crimes", where you show that your method can invert data from your own forward model. Being able to invert simulated data from your own forward model is a necessary, but far from sufficient, criterion to having a useful working method. Thus, while figure 5 and the associated table are useful in understanding some of the robustness of our approach, they are not sufficient. This is why we (on the fourth line of table 1) specifically evaluate our method on raw experimental data, not simulated data. This is data that our approximated forward model has never seen, and our inverse model has never seen. This reflects the most realistic real-world application of our approach to unseen data.

In the camera-ready manuscript, we will attempt to make the exact evaluation role of each row in Table 1 more clear. In particular, the first row is an "inverse crime" – we are evaluating our ability to invert spectra that both our forward model and our inverse model have seen during training. We do this to show how well our model could work in the best case scenario, and to provide some context for understanding the relative accuracy of the other rows.

Reviewer three also pointed out that we do not define AUC in our paper, an omission which we will correct in the camera ready. Our method ultimately calculates the difference between the (predicted) spectrum of a candidate structure with the experimentally-observed spectrum. The difference (in $\ell_2$) between these spectra allows us to set an error threshold, below which we will consider the structure to be "correct". We can compute the area under the curve that arises from varying this threshold, thus giving rise to the AUC. To reviewer one's comment that the AUCs seem low: we find that most users of our method will want a high degree of confidence that the recovered structure is correct, and will likely set this threshold very conservatively, thus making AUC a less useful assessment of overall performance. For predicted structures above the threshold, the user can then discard the predictions – our algorithm simply could not find a sufficiently good structure. This is why we have the other columns in table 1, which should be interpreted as follows: if you set the threshold such that you predict a candidate structure N% of the time, how correct are those structures? We see that in the real-world use case (row 4), if you set the threshold so conservatively that our method only predicts a structure 20% of the time, 97.1% of the those returned structures are correct.

Finally, our fast forward model is described in another publication that is currently in press, and thus we glossed over it in this paper. It is a fairly straight-forward application of a graph convolutional network to predict per-vertex properties (in this case, the chemical shift values) with some minor extensions for incorporating uncertainty. In our camera ready we will add several paragraphs to the supplemental section more fully describing this model, as well as linking to the paper describing this approach.

All reviewers asked if the resulting learned inverse model was interpretable, or if the learned features captured the intuition that humans use when solving this problem. Preliminary experiments looked at which bonds and bond types were placed with which nuclei, and suggest the answer is yes. Initially higher-order bonds (double and triple) are often placed first in sequence by our approach, as they are often associated with the most unique (highest entropy) observed spectral values. We will include the results of these analysis in the supplemental section of the camera-ready.

Reviewer one asked about network architecture; we made a reasonable (although not exhaustive) effort to identify a good architecture, considering many architectures derived in the graph prediction literature. In this context, the referenced Battaglia paper was of considerable help, as their "relational network" formulation unified these approaches and allowed us to zero-in on a reasonable set of hyperparameters, allowing us to incorporate per-vertex measurements, sparse per-edge observations (the edges of the current state), and yet output a dense probability distribution over subsequent edges. We have not considered a transformer-like architecture thus far but may in the future, given their success in other structured prediction (NLP) problems.

Reviewer asked for clarification of "Beta" on line 3; when we are generating training data we delete edges in the source molecule with a probability $p$ where $p$ is sampled from a mixture of a uniform and a beta distribution,

$$p \sim \frac{3}{10}\text{Unif}(0,1) + \frac{7}{10}\text{Beta}(3,3)$$

The beta component guarantees we sample a lot of "partially completed" molecules for training. For these sorts of sequential structured prediction problems, "distributional shift" is often a challenge, but we found empirically that by over-sampling partially completed molecules according to this distribution we achieved superior reconstruction performance compared to a mere uniform sample. The investigation of active-learning approaches (like DAGGAR and SEARN ) is another way and represents an avenue for exploration in the future.

We thank the reviewers again for their helpful and constructive feedback.

[Meta-Review · NeurIPS 2019]

The paper studies a problem of predicting the molecular structured given its NMR spectrum and the molecular formula, through deep imitation learning. The reviewers find the topic important for cheminformatics and the proposed method relevant and potentially impactful. The write-up of the paper should be improved.